# *In Vitro* Effects of Pesticides on European Foulbrood in Honeybee Larvae

**DOI:** 10.3390/insects11040252

**Published:** 2020-04-17

**Authors:** Sarah C. Wood, Jocelyne C. Chalifour, Ivanna V. Kozii, Igor Medici de Mattos, Colby D. Klein, Michael W. Zabrodski, Igor Moshynskyy, M. Marta Guarna, Patricia Wolf Veiga, Tasha Epp, Elemir Simko

**Affiliations:** 1Department of Veterinary Pathology, Western College of Veterinary Medicine, University of Saskatchewan, 52 Campus Drive, Saskatoon, SK S7N 5B4, Canada; jocelyne.chalifour@usask.ca (J.C.C.); ivanna.kozii@usask.ca (I.V.K.); igor.mattos@usask.ca (I.M.d.M.); colby.klein@usask.ca (C.D.K.); michael.zabrodski@usask.ca (M.W.Z.); igm800@mail.usask.ca (I.M.); elemir.simko@usask.ca (E.S.); 2Beaverlodge Research Farm, Agriculture and Agri-Food Canada, 1 Research Road, Beaverlodge, AB T0H 0C0, Canada; marta.guarna@canada.ca; 3National Bee Diagnostic Centre, Grand Prairie Regional College, 1 Research Road, Beaverlodge, AB T0H 0C0, Canada; PWolfVeiga@GPRC.ab.ca; 4Department of Large Animal Clinical Sciences, Western College of Veterinary Medicine, University of Saskatchewan, 52 Campus Drive, Saskatoon, SK S7N 5B4, Canada; tasha.epp@usask.ca

**Keywords:** European foulbrood, *Melissococcus plutonius*, atypical, honeybee, neonicotinoid, fungicide, thiamethoxam, boscalid, pyrimethanil, propiconazole

## Abstract

Neonicotinoid and fungicide exposure has been linked to immunosuppression and increased susceptibility to disease in honeybees (*Apis mellifera*). European foulbrood, caused by the bacterium *Melissococcus plutonius*, is a disease of honeybee larvae which causes economic hardship for commercial beekeepers, in particular those whose colonies pollinate blueberries. We report for the first time in Canada, an atypical variant of *M. plutonius* isolated from a blueberry-pollinating colony. With this isolate, we used an *in vitro* larval infection system to study the effects of pesticide exposure on the development of European foulbrood disease. Pesticide doses tested were excessive (thiamethoxam and pyrimethanil) or maximal field-relevant (propiconazole and boscalid). We found that chronic exposure to the combination of thiamethoxam and propiconazole significantly decreased the survival of larvae infected with *M. plutonius*, while larvae chronically exposed to thiamethoxam and/or boscalid or pyrimethanil did not experience significant increases in mortality from *M. plutonius* infection *in vitro*. Based on these results, individual, calculated field-realistic residues of thiamethoxam and/or boscalid or pyrimethanil are unlikely to increase mortality from European foulbrood disease in honeybee worker brood, while the effects of field-relevant exposure to thiamethoxam and propiconazole on larval mortality from European foulbrood warrant further study.

## 1. Introduction

European foulbrood (EFB), caused by the bacterium *Melissococcus plutonius*, is an enteric disease of honeybee (*Apis mellifera*) larvae [1]. *M. plutonius* is transmitted to developing larvae through contaminated brood food and proliferates within the larval midgut, leading to larval death, especially under conditions of colony stress [1]. Honeybee larvae respond to bacterial infection through both cellular [2] and humoral immunity [3], provided by hemocytes and antimicrobial peptides (AMPs), respectively.

Strains of *M. plutonius* have been categorized as ‘atypical’ variants and ‘typical’ variants which have genetic differences [4], including variation in cell-adhesion proteins and carbohydrate metabolism [5]; different virulence factors [6]; biochemical differences, including variation in β-glucosidase activity, esculin hydrolysis, and carbohydrate fermentation [7]; as well as variable fastidiousness in their requirements for successful growth in culture media [7]. The *in vitro* infection of honeybee worker larvae with *M. plutonius* has been successfully performed [7,8,9], with atypical strains of *M. plutonius* showing higher incidence of larval mortality in comparison with typical strains [7,8,9,10,11]. 

There is a widespread, chronic, in-hive exposure of honeybees to complex mixtures of agricultural and apicultural fungicides and insecticides through nectar, honeydew, honey, wax, pollen and pollen stored as beebread [12,13,14]. Moreover, 98.4% of pollen and wax was found to have two or more pesticide residues and 61.7% of pollen or wax containing a fungicide also contained insecticide or miticide residues [12]. Increased numbers of pesticide residues in wax, particularly fungicides, which inhibit sterol biosynthesis [15], have been significantly associated with colony mortality [13]. There is also a concern for the negative effects of chronic, in-hive pesticide exposure on the developing worker brood [16], although the transfer of pesticides from pollen and honey to royal jelly is considered to be low, ranging from 0.001%–0.016% [17].

Concentrations of insecticides and fungicides within hive matrices are generally considered to be sublethal for honeybees [12]. In-hive pesticide surveillance in Europe, Asia, and North and South America [18] detected the fungicide boscalid (BOS) in 12.6% of wax and 4.3% of pollen, at means of 72.4 ng/g and 22.5 ng/g respectively; the fungicide pyrimethanil (PYR) in 1.4% of wax and 3.5% of pollen at means of 14.3 ng/g and 14.2 ng/g, respectively; the fungicide propiconazole (PROP) in 1% of wax and 1.8% of pollen at means of 196.5 ng/g and 5.5 ng/g, respectively; and the neonicotinoid insecticide thiamethoxam (THI) in 7.7% of wax, 12.8% of pollen, and 65% of honey at means of 38 ng/g, 28.9 ng/g, and 6.4 ng/g, respectively. The mean concentration of THI in honey has been reported to be as high as 17.2 ng/g in Saskatchewan, Canada [19], but globally, the average THI concentration in honey has been calculated to be 0.29 ng/g [20]. The environmental concentrations of THI in pollen and nectar have been reported as being as high as 86 ng/g in pollen from wildflowers adjacent to oilseed rape grown from THI-treated seed [21], and as high as 13.3 ng/g in the nectar of oilseed rape grown from THI-treated seed [21]. By comparison, the adverse effects of THI exposure on honeybee colonies are not observed until THI concentrations reach 20 to 100 ng/g [22,23,24,25].

Chronic co-exposure of honeybees to fungicides and insecticides within a colony has the potential for synergistic negative effects on honeybee health. Compared to other insects, honeybees have fewer genes encoding cytochrome P450 monooxygenase (P450) enzymes used in pesticide detoxification [26], and some fungicides, such as PROP, are inhibitors of insect P450s [15]. Not surprisingly, the laboratory co-exposure of honeybees to PROP and insecticides has been shown to synergistically increase toxicity and decrease survival of honeybee adult workers [27,28,29] and worker larvae [30].

Neonicotinoid and fungicide exposure may also alter honeybee susceptibility to pathogens through changes in innate immune function [31,32] and social immunity [33]. Neonicotinoids are hypothesized to immunosuppress honeybees by the downregulation of immune genes and pathways [34], including transcription factor NF-κB [35]. For example, the neonicotinoid clothianidin was shown to decrease cellular immunity and increase the mortality of larvae infected with bacterial spores of *Paenibacillus larvae*, the etiologic agent of American foulbrood [2]. Furthermore, fungicide exposure in pollen increased the risk of the laboratory infection of adult honeybee workers with the microsporidian parasite *Nosema ceranae*; however, neonicotinoid exposure was associated with decreased *Nosema* infection prevalence in honeybee workers [36].

Beekeepers pollinating blueberries in Canada and the United States have reported an increased incidence of EFB during and after pollination [37,38,39,40]. Elevated levels of fungicides in beebread from blueberry pollination were significantly correlated with colony loss [13]. However, the relationship between fungicide exposure during pollination and EFB is unknown. 

To date, no one has investigated whether pesticide exposure alters the susceptibility of honeybee worker larvae to EFB. Thus, we used an *in vitro* model to test the hypothesis that pesticide exposure increases the mortality of worker honeybee larvae from EFB. Specifically, we determined whether or not honeybee worker larvae are more susceptible to EFB-associated mortality when exposed to (i) the insecticide THI, (ii) the fungicides BOS, PROP or PYR, or (iii) the combination of THI and BOS, PROP or PYR.

## 2. Materials and Methods 

To investigate the effects of pesticide exposure on honeybee larval mortality from EFB *in vitro*, newly hatched worker larvae were infected with a pure culture of *M. plutonius* on the day of grafting (day 0 (D0)) and exposed to pesticides in the diet from D0 to D5. The survival of the larvae from D0 to D6 was compared between the pesticide-exposed larvae and controls (Table 1).

### 2.1. Isolation of an Atypical Variant of M. Plutonius 

*M. plutonius* was isolated from a diseased larva from a honeybee colony in blueberry pollination in the Fraser Valley of British Columbia, Canada. Briefly, the macerated larva was streaked on KSBHI agar [brain heart infusion (Difco; Becton, Dickinson and Co., Sparks, MD, USA) media with 0.15 M KH_2_PO_4_ (Millipore Sigma, Oakville, ON, Canada), 1% soluble starch (Difco; Becton, Dickinson and Co.), 1.5% agar (Difco; Becton, Dickinson and Co.) and 3 µg/mL filter-sterilized nalidixic acid (Millipore Sigma)] [7,41] and incubated at 37 °C for 3 days under microaerophilic conditions (Pack-MicroAero, Mitsubishi Gas Chemical Co. Inc., Tokyo, Japan).

Colonies resembling *M. plutonius* [41] were subcultured on KSBHI agar and *M. plutonius*’ identity was confirmed using a Gram stain (Appendix A) and PCR (Appendix A) [42]. Duplex PCR (Appendix A) [4] identified the *M. plutonius* isolate as an atypical variant, which was further characterized using multi-locus sequence typing (MLST) [43]. Based on the comparison to the *M. plutonius* MLST databases (https://pubmlst.org/mplutonius/) [44], the isolate belonged to sequence type (ST) 19 of Clonal Complex (CC) 12, which was previously identified in the Netherlands [43]. The GenBank accession numbers for the MLST loci sequenced for our *M. plutonius* isolate are as follows: MT127566, MT127567, MT127568, MT127569, MT127570, MT127571 and MT127572. 

The *M. plutonius* isolate was subcultured in liquid KSBHI media [7] and incubated at 37 °C under microaerophilic conditions and shaking at 100 rpm, to an optical density at 600 nm (OD_600_) of 1.6. The culture was mixed with 20% glycerol and stored in 150 µL aliquots at −80 °C, which served as the stock culture for all experiments. 

### 2.2. Experimental Animals 

From mid-June through mid-August, 2019, synchronized frames of worker larvae were continuously generated from six caged queens within experimental colonies of *A. mellifera*, located at the University of Saskatchewan (Saskatoon, Saskatchewan, Canada). Every 24 h, a frame of freshly-laid eggs was removed from the cage in each colony and replaced with an empty frame of foundation drawn with wax. The frames of eggs were incubated within the colony for three days until hatching, after which the frames were transported to the laboratory in a portable incubator at 35 °C.

### 2.3. In Vitro Larval Rearing 

*A. mellifera* worker larvae were reared *in vitro* [45] for six days (Table 1). Larval diets ‘A’, ‘B’, and ‘C’ [45] were prepared from royal jelly (Stakich, Troy, MI, USA), D-glucose (Fisher Scientific, Toronto, ON, Canada), fructose (Fisher Scientific), yeast extract (Fisher Scientific), and distilled water. The larvae were fed diets ‘A’, ‘B’, and ‘C’ in sequence [45] and the three diets differed in the proportion of ingredients they contained to reflect the changes in worker diet composition (gradual increase in sugar and protein with increasing age of larva), fed to worker larvae in a colony [45]. Diet aliquots were frozen at −20 °C and warmed to 35 °C prior to feeding. 

On D0, within a sterilized biosafety cabinet, newly-hatched, first instar larvae were grafted from their frame into sterilized queen cell cups (Apihex, Calgary, AB, Canada), primed with 10 µL of control diet ‘A’, within a 48-well sterile tissue culture plate (STCP) (Fisher Scientific), kept on a warming pad at 35 °C [45]. Each STCP received larvae from two to three different genetic lineages (16–24 larvae per lineage). After grafting, each larva was fed an additional 9.5 µL of control or treatment diet ‘A’, mixed with 0.5 µL of sterile phosphate buffered saline (PBS), or 0.5 µL of *M. plutonius* culture diluted in sterile PBS (see Section 2.5 below and Table 1). After feeding, the STCPs were incubated in one of two desiccators containing supersaturated K_2_SO_4_ solution (Fisher Scientific) [45], within an incubator set at 35 °C. From D2 to D5, larvae received daily feedings of control or treatment diet ‘B’ or ‘C’, according to the feeding schedule of Schmehl et al. (2016) [45] From D1 to D6, larvae were examined daily and dead larvae were identified based on their discolored, deflated appearance and lack of moving spiracles [46] using unaided visual examination and/or a stereomicroscope (Figure 1). Dead larvae were removed each day and recorded.

We had two types of negative controls: a ‘survival’ control (no pesticide and no *M. plutonius*), which was required to have ≥75% survival at D6 for the data from the corresponding STCP to be included in the study; and an ‘infected’ control (no pesticide and infected with 500, 250, or 50 CFU *M. plutonius*), for comparison to pesticide-exposed larvae, which were infected with the same number of CFU of *M. plutonius* (Table 1). Each STCP was divided into four groups including a survival control group on every plate, with 10–12 larvae per group (mean = 11.94, SD = 0.26). Pesticide-treated larvae received pesticides only, or pesticides in combination with *M. plutonius* (Table 1). Four to seven replicates (mean = 53.05 larvae, SD = 9.02) of each treatment and control group were performed, with the exception of the survival control, which had 16–23 replicates (mean = 236.5 larvae, SD = 46.65). Replicates of each group were performed on a minimum of two different, time-staggered STCPs. 

Temperature and relative humidity in each of the two desiccators were logged hourly using a HOBO MX Temp/RH Data Logger MX1101 (Onset Computer Corp., Bourne, MA, USA) and found to be, on average, 34.69 °C (SD = 0.26) and 93.11% (SD = 10) and 34.71 °C (SD = 0.18) and 96.76% (SD = 6.76), respectively, for the duration the experiment. 

### 2.4. Larval Pesticide Exposure

Pesticide stock solutions were prepared in water and/or acetone from analytical standard chemicals (Millipore Sigma). THI (Product 37924, Lot BCBT8326, expiry March 2022) was prepared as a 100 ng/µL stock in distilled water. PROP (Product 45642, Lot BCBW6694, expiry February 2023) was prepared as a 65 ng/µL stock in distilled water and 0.0065% acetone. PYR (Product 31577, Lot BCBW1407, expiry November 2022) was prepared as a 65 ng/µL stock solution in distilled water and 0.5% acetone. Two stock solutions of BOS were prepared. BOS (Product 33875, Lot BCB58868V, expiry Aug 2021) was prepared as a 1170 ng/µL stock in 100% acetone and BOS (Product 33875, Lot SZBF099XV, expiry April 2020) was prepared as a 1600 ng/µL stock in 100% acetone. 

The pesticide concentration remained constant throughout the experiment. Since larvae were grafted into queen cell cups primed with non-contaminated diet ‘A’ on D0, the pesticide concentration in diet ‘A’ was adjusted accordingly, to account for this dilution. For the larvae treated with THI and/or PROP or PYR, part of the water in the diet was replaced with pesticide stock solution, to achieve the desired pesticide concentration in the larval diet (Table 2). Due to its poor water solubility, BOS was added by pipetting BOS stock solution (representing 2.5% of diet volume), into the diet adjacent to each larva after feeding. Survival controls for the BOS-treated larvae received an equal volume (2.5%) of acetone. As a positive control (Table 1), to confirm BOS activity and exposure, 4 replicates (48 larvae) of 40 ng/µL BOS with 10 ng/µL THI (2.5% acetone) and 4 replicates (48 larvae) of 80 ng/µL BOS with 10 ng/µL THI (5% acetone) were performed, and survival was compared to survival control larvae treated with 2.5% acetone (47 larvae) or 5% acetone (48 larvae), respectively. 

The THI concentrations selected (Table 2) were based on previously tested THI concentrations which did not significantly impact larval survival [16], and the total dose of THI provided in the diet represented 84.2–842 times the calculated, worst-case, field-relevant exposure of a worker larva to THI (1.9 ng), which was calculated based on estimated worker larval consumption of 5.4 mg pollen and 180 mg nectar during development [47], and reported maximal environmental concentrations of THI in pollen (86 ng/g) and nectar (13.3 ng/g) [21]. Thus, the concentrations of THI tested were not intended to be field-realistic; but instead, high concentrations with the potential for observable, sublethal effects. The total doses of PROP and BOS administered in the diet (Table 2) were based on previously tested [30], field-relevant doses of PROP and BOS calculated based on maximum application rates of these fungicides to almond crops [30]. We used the same total dose of PYR as for PROP (Table 2). According to the maximum residues of PYR reported in pollen (83 ng/g) and nectar (4 ng/g) [18], the total dose of PYR tested in our experiment represents 2876 times the calculated, worst-case, field-relevant exposure of a worker larva to PYR (0.779 ng) [47].

### 2.5. Larval Infection with M. Plutonius 

For experimental larval infection, fresh cultures of *M. plutonius* were prepared daily by thawing an aliquot of stock culture (see Section 2.1 above), diluting it 1/1000 in liquid KSBHI media, and growing the culture for 29 h at 37 °C under microaerophilic conditions, and shaking at 100 rpm. To determine the bacterial load of the stock culture after 29 h, the OD_600_ of the culture was measured and serial dilutions were prepared in sterile PBS. Serial dilutions were plated on KSBHI agar and incubated at 37 °C for 3 days under microaerophilic conditions, to determine colony forming units (CFU) per ml. 

The mean OD_600_ of the cultures used for experimental infection was 0.849 (SD = 0.206) and the mean CFU/mL based on the plating of serial dilutions was 9.58 × 10^7^ CFU/mL (SD = 2.85 × 10^7^). On the day of grafting, each larva was administered 0.5 µL of a 1/100, 1/200, or 1/1000 dilution of the stock culture; thus, each larva received 479 CFU (SD = 142.46), 240 CFU (SD = 71.23), or 47.9 (SD = 14.25) CFU, respectively, based on the mean CFU/mL of the stock culture. For simplicity, the bacterial inocula will be referred to as 500, 250, or 50 CFU. Larvae in the survival control group received 0.5 µL of sterile PBS (Table 1). 

To verify fulfillment of Koch’s postulates, control and *M. plutonius*-infected larvae were preserved in 10% neutral phosphate buffered formalin, processed for histopathology using standard automatic tissue processing, embedded in paraffin (Paraplast Plus, Leica Biosystems, Richmond, IL, USA), sectioned into 5 µm sections, and stained with Hematoxylin and Eosin and/or Gram stain [51,52]. Control and *M. plutonius*-infected larvae were also homogenized in sterile PBS, streaked on KSBHI agar and incubated at 37 °C for 3 days under microaerophilic conditions. 

### 2.6. Statistical Analysis 

A statistical analysis was conducted using Stata/SE 16 (College Station, TX, USA). For each dose of *M. plutonius* (0, 50, 250, 500 CFU), the larval survival at D6 was compared between pesticide-treated and control groups, using a Pearson Chi-squared test. For the dose of *M. plutonius* resulting in the approximately 50% survival of infected controls, an additional survival analysis was performed using Cox proportional hazards regression, with a post hoc global test to confirm the proportionality of hazards.

## 3. Results

We successfully isolated an atypical strain of *M. plutonius* and reproduced EFB disease *in vitro* (Figure 1). We also fulfilled Koch’s postulates by demonstrating gram positive bacteria within the midgut of infected larvae on histopathology (Figure 1d) and culturing back *M. plutonius* from infected larvae. Control larvae did not contain bacteria within the midgut on histopathology (Figure 1c), and the culture of control larvae did not yield colonies of *M. plutonius*. 

Chronic exposure to THI in combination with PROP significantly decreased the survival of larvae infected with 50 CFU *M. plutonius*
*in vitro* (Figure 2d and Figure 3d). Compared to infected control larvae which received 50 CFU *M. plutonius*, larvae exposed to 1 ng/µL THI and 14 ng/µL PROP and infected with 50 CFU *M. plutonius* had a significantly lower (by 25%) survival over 6 days (Figure 2d, *X^2^*(1) = 3.9625, *p* = 0.047). Similarly, when larval survival after infection with 50 CFU *M. plutonius* was analyzed using Cox proportional hazard regression (Figure 3), we observed a marginally significant decrease in the survival of larvae exposed to THI and PROP compared to infected controls (Figure 3d; *p* = 0.048, Hazard Ratio = 1.85, 95% Confidence Interval (CI) = 1.00 to 3.42; Appendix A). 

By comparison, chronic larval exposure to THI, BOS, PYR, PROP, or THI in combination with the fungicides BOS or PYR, was not shown to significantly affect larval survival over 6 days after infection with 50 CFU *M. plutonius* (Figure 2 and Figure 3; Appendix A). Similarly, at higher doses (250 and 500 CFU) of *M. plutonius*, there was no significant effect of THI and/or fungicide treatment on survival relative to infected controls (Figure 2). 

In the absence of *M. plutonius* infection, we observed a significant, 9% and 8%, respectively, lower survival of larvae exposed to PROP (Figure 2d, *X^2^*(1) = 6.095, *p* = 0.014) and THI with PROP (*X^2^*(1) = 5.88, *p* = 0.015), compared to controls. Uninfected larvae, which were chronically exposed to THI, BOS, PYR, or THI in combination with BOS or PYR, did not experience a significant (*p* > 0.05) decrease in larval survival relative to survival controls (Figure 2a–c).

Positive control, high concentrations of 40 and 80 ng/µL BOS with 10 ng/µL THI significantly decreased larval survival relative to survival controls by 32.19% (*X^2^*(1) = 8.1934, *p* = 0.004) and 96.28% (*X^2^*(1) = 28.6138, *p* < 0.001), respectively, in the absence of *M. plutonius*, confirming the efficacy of the pesticide exposure model used for BOS. 

## 4. Discussion

We report an atypical isolate of *M. plutonius* for the first time in Canada. The distribution of atypical isolates in Canada is currently unknown. With this isolate, we successfully reproduced EFB disease *in vitro* (Figure 1) and using this *in vitro* model, we demonstrated that chronic exposure to a neonicotinoid (THI) or one of three fungicides (BOS, PYR, or PROP) on its own does not increase honeybee worker larvae death from EFB (Figure 2 and Figure 3). However, chronic co-exposure of worker larvae to THI and PROP was shown to significantly decrease the survival of larvae infected with *M. plutonius*, relative to infected controls (Figure 2d and Figure 3d). We reiterate that only one of four combinations of THI with a fungicide tested was correlated with a significant increase in mortality from EFB, and the dose of THI used was 84.2 times greater than environmentally relevant exposure; thus, our study does not show that pesticide exposure would predispose honeybees to EFB-associated mortality in the field. 

After infection with 50 CFU *M. plutonius*, the survival of THI and PROP-exposed larvae over six days was significantly lower than infected controls (Figure 2d), with chronic THI and PROP exposure significantly increasing the risk of larval mortality by 1.85 times (*p* = 0.048, Figure 3d, Appendix A), relative to infected controls receiving 50 CFU of *M. plutonius*. This finding suggests that THI and PROP co-exposure may have potentiated development of EFB disease, possibly through the PROP-mediated inhibition of larval P450s, leading to decreased THI detoxification and the subsequent THI-mediated impairment of larval antibacterial defenses. The hazard of 1 ng/µL THI with PROP exposure after infection with 50 CFU *M. plutonius* (Figure 3d, Appendix A) was greater than the hazard of exposure to 1 ng/µL THI on its own (Figure 3a), which resulted in a non-significant, 1.52 times increase (*p* = 0.17, Appendix A) in larval mortality, relative to infected controls administered 50 CFU of *M. plutonius*. There is previous evidence for the PROP-mediated inhibition of honeybee P450s, leading to increases in the toxicity of THI to adult workers [29] and decreases in the survival of worker larvae exposed to the diamide insecticide chlorantraniliprole [30]. The absence of significant differences in survival relative to infected controls of larvae infected with 250 and 500 CFU of *M. plutonius* and exposed 1 ng/µL THI with PROP (Figure 2d) could be due to these higher doses of *M. plutonius* overwhelming the larval immune system, regardless of its immunocompetence. As well, we emphasize that larval survival was only monitored over six days in our experiment, and we cannot rule out a possible time lag in mortality of our infected control group. 

Furthermore, consistent with our results, other studies [2,53] have demonstrated the increased susceptibility of honeybee larvae to disease when exposed to agrochemicals. For example, the neonicotinoid clothianidin and the bacterium *Paenibacillus larvae* were shown to act synergistically to decrease survival and total hemocyte count of exposed larvae [2]. Similarly, larvae infected with four RNA viruses and chronically exposed to an organosilicone surfactant adjuvant used in tank mixes of pesticides had significant increases in mortality and viral replication and decreased expression of Toll 7-like receptor which mediates viral immunity [53]. One criticism of these studies [2,53], as well as the present study, is that all experiments lacked a control group of agrochemical-exposed larvae which were infected with a non-pathogenic organism which was similar to the pathogen under study. 

Alternatively, the significant decrease in survival of the larvae co-exposed to THI and PROP and infected with 50 CFU of *M. plutonius* (Figure 2d) could be explained by the direct toxic effects of PROP, rather than the increased susceptibility to *M. plutonius*, considering that chronic larval exposure to PROP, or THI with PROP, in the absence of *M. plutonius*, resulted in significant 9% (*p* = 0.014) and 8% (*p* = 0.015), respectively, decreases in larval survival relative to survival controls, although no significant effect of PROP exposure on its own was observed in the presence of *M. plutonius* (Figure 2d). The total PROP dose (2.24 µg) administered in our study was based on maximum field application rates to almonds [30]. Wade et al. (2019) [30] found no significant effect of 2.25 µg PROP on the survival of larvae exposed on day 4 of development, but perhaps the chronic exposure scenario in our study provided more time for the negative effects of PROP on larval survival to occur. Considering that the total THI dose (160 ng), tested alone and in combination with PROP, was 84.2 times higher than the calculated, maximum environmental exposure of worker larvae [47], and on its own THI did not show significant effects on larval survival, further studies are needed to confirm field-relevant doses and to examine the effect of these doses of THI in combination with PROP on larval mortality from *M. plutonius*. 

In contrast to studies such as ours which demonstrate negative effects, or no effect of pesticides on susceptibility of larval honeybees to infectious disease, there is some evidence to suggest that pesticides may have a positive, immunostimulatory effect on honeybees. For example, the *in vitro* fungicide exposure of honeybee larvae was found to increase gene expression of an immune enzyme in pupae involved in melanization [54]. Additionally, Dickel et al. (2018) [55] observed a possible hormetic effect of the neonicotinoid thiacloprid on survival of adult workers co-exposed to the bacterium *Enterococcus faecalis*, suggesting that concurrent bacterial infection and sublethal pesticide exposure may increase the longevity of adult honeybees. Future studies with our *in vitro* model of EFB should examine the effect of pesticides on worker mortality over the entire developmental period to eclosion, as well as sublethal parameters, such as bacterial load or immune gene expression.

The relevance of the results reported herein is limited to the single atypical *M. plutionius* isolate we tested, and cannot be generalized to other isolates of *M. plutonius* without additional *in vitro* testing. Of note, it is interesting that we reliably reproduced EFB disease *in vitro* with only 50 CFU of *M. plutonius*, while other authors required 56 [9] to 1000 [7] times greater infectious doses of atypical *M. plutonius* to trigger EFB *in vitro*. Differences in strain virulence may explain the discrepancy in these infectious doses. 

## 5. Conclusions

An *in vitro* model for testing the effects of pesticide exposure on the development of the EFB disease in honeybee larvae was successfully implemented with an atypical isolate of *M. plutonius* from a blueberry-pollinating colony. Using this model, we demonstrated that a neonicotinoid insecticide (thiamethoxam) and/or the fungicides boscalid or pyrimethanil do not increase the susceptibility of worker honeybee larvae to mortality from EFB. However, chronic exposure to greater than field-realistic concentrations of thiamethoxam with the fungicide propiconazole were shown to significantly increase larval mortality from EFB at low infectious doses *in vitro*, suggesting that further testing of field-relevant thiamethoxam concentrations in combination with propiconazole is required. Our established experimental model will enable future testing of additional pesticide combinations to better understand the interaction between pesticides and larval susceptibility to EFB. Studies such as this are important to strike a balance between the farmers’ need to control crop pests with agrochemicals and the beekeepers’ need for healthy colonies with which to provide pollination services and produce honey.

## Figures and Tables

**Figure 1 insects-11-00252-f001:**
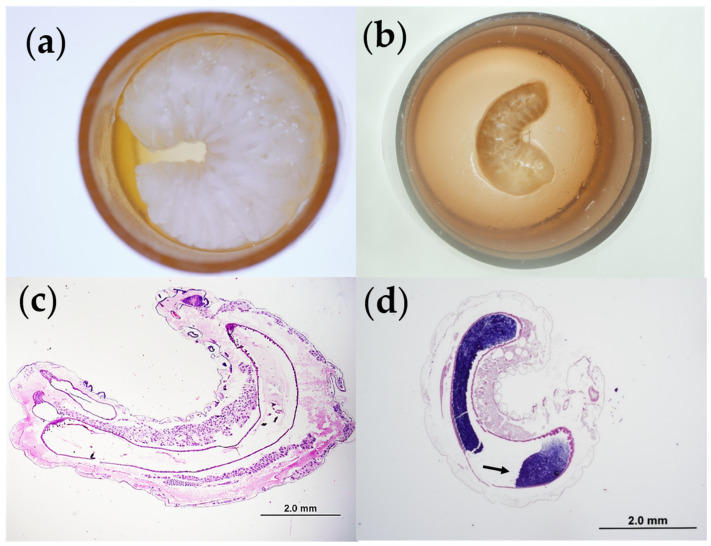
*In vitro* multiplication of *Melissococcus plutonius* in honeybee worker larvae. Gross (**a**,**b**) and histologic sections (**c**,**d**) after 6 days of *in vitro* rearing of control larvae (**a**,**c**) and larvae infected with *M. plutonius* (**b**,**d**). The healthy control larvae (**a**) is white and plump compared to the larvae infected with *M. plutonius* (**b**), which is decreased in mass, brown and deflated, with prominent tracheae. The Gram-stained section of an infected larva (**d**) demonstrates a mass of gram-positive bacteria (arrow) within the midgut which is absent in the section of a control larva stained with Hematoxylin and Eosin (**c**).

**Figure 2 insects-11-00252-f002:**
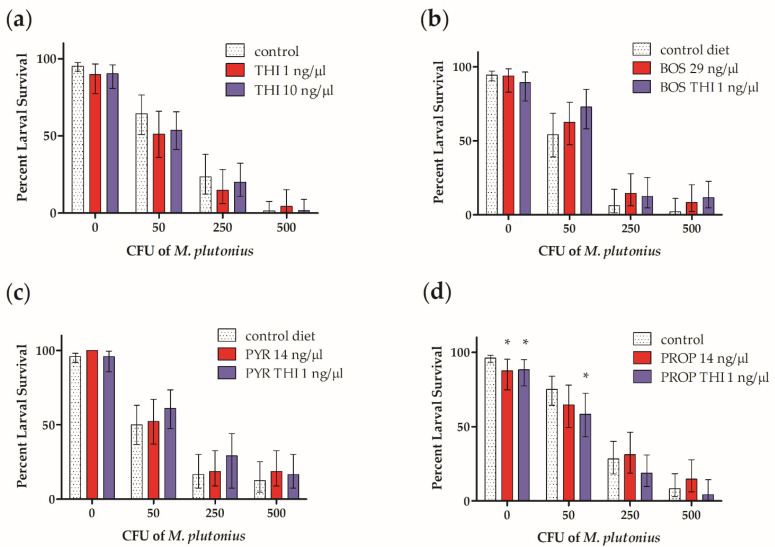
Effects of chronic pesticide exposure on larval survival from European foulbrood. (**a**) Percentage survival of larvae fed control diet or diet with thiamethoxam (THI; 1 or 10 ng/µL); (**b**) Percentage survival of larvae fed control diet, diet with boscalid (BOS; 29 ng/µL), or diet with BOS and THI (1 ng/µL); (**c**) Percent survival of larvae fed control diet, diet with pyrimethanil (PYR; 14 ng/µL), or diet with PYR and THI (1 ng/µL); (**d**) Percentage survival of larvae fed control diet, diet with propiconazole (PROP; 14 ng/µL), or diet with PROP and THI (1 ng/µL). Bars show percent larval survival at day 6 with 95% confidence interval for 45–84 worker honeybee larvae reared *in vitro* and infected with 0, 50, 250, or 500 colony forming units (CFU) of *Melissococcus plutonius* and 191–300 survival control larvae, which were unexposed to pesticides and not infected with *M. plutonius*. Percentage larval survival was analyzed with a Chi-squared test. * indicates significant difference (*p* < 0.05) relative to control for each inoculum (CFU) of *M. plutonius*. Larval survival from European foulbrood was significantly decreased by co-exposure to the insecticide thiamethoxam with the fungicide propiconazole.

**Figure 3 insects-11-00252-f003:**
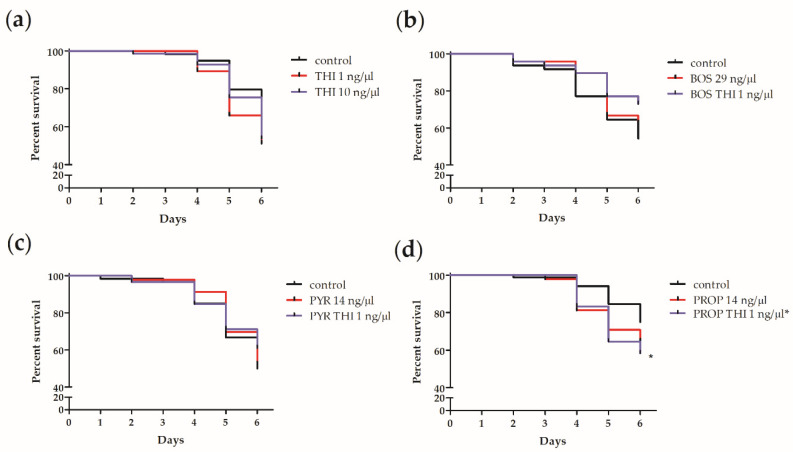
Effects of chronic pesticide exposure on survival of larvae infected with 50 colony forming units (CFU) of *Melissococcus plutonius*. (**a**) Percentage survival of larvae fed control diet or diet with thiamethoxam (THI; 1 or 10 ng/µL); (**b**) Percentage survival of larvae fed control diet, diet with boscalid (BOS; 29 ng/µL), or diet with BOS and THI (1 ng/µL); (**c**) Percentage survival of larvae fed control diet, diet with pyrimethanil (PYR; 14 ng/µL), or diet with PYR and THI (1 ng/µL); (**d**) Percentage survival of larvae fed control diet, diet with propiconazole (PROP; 14 ng/µL), or diet with PROP and THI (1 ng/µL). From day 0 to day 5, larvae were fed control or pesticide-contaminated diet and mortality was recorded daily for 6 days after grafting on day 0. Lines indicate percent daily survival for 46–84 larvae administered 50 CFU on day 0 of *in vitro* rearing. * indicates significant (*p* < 0.05) difference relative to control by Cox proportional hazards regression. Thiamethoxam and propiconazole exposure increased susceptibility of honeybee worker larvae to mortality from European foulbrood *in vitro*, after infection with 50 CFU of *Melissococcus plutonius*.

**Table 1 insects-11-00252-t001:** Experimental design of *in vitro* model for testing effects of pesticides on larval mortality from European foulbrood. On day 0 (D0) of the experiment, larvae received 0.5 µL phosphate buffered saline (PBS) or 0.5 µL of a pure culture of *Melissococcus plutonius* diluted in PBS to contain 500, 250, or 50 colony forming units (CFU). From day 0 to day 5, larvae were administered control diet or diet contaminated with the pesticides thiamethoxam (THI) and/or boscalid (BOS), pyrimethanil (PYR), or propiconazole (PROP). Larval survival was monitored daily until day 6.

Experimental Group	Inoculation with *M. Plutonius* D0	Pesticide Administration D0 to D5
Pesticide and *M. plutonius*	0.5 µL *M. plutonius* with 500, 250, or 50 CFU	THI and/or BOS, PYR, or PROP
Pesticide only	0.5 µL PBS	THI and/or BOS, PYR, or PROP
Survival control	0.5 µL PBS	none
Infected control	0.5 µL *M. plutonius* with 500, 250, or 50 CFU	none
Positive control	0.5 µL PBS	THI and BOS

**Table 2 insects-11-00252-t002:** Pesticides, mode of action, concentration and total dose in 160 µL larval diet, provided from day 0 to day 5, to honeybee worker larvae reared *in vitro*.

Pesticide	Mode of Action	Diet Concentration (ng/µL)	Total Dose (ng)
Thiamethoxam (THI)	Neonicotinoid insecticide which is a nicotinic acetylcholine receptor agonist [48]	1, 10	160, 1600
Boscalid (BOS) ^†^	Carboxamide fungicide which inhibits cellular respiration [49]	29	4680
Pyrimethanil (PYR)	Anilinopyrimidine fungicide which inhibits protein synthesis [50]	14	2240
Propiconazole (PROP)	Triazole fungicide which inhibits sterol biosynthesis and cytochrome P450 monooxygenase enzymes [15]	14	2240

† Due to its poor water solubility, BOS was pipetted into the larval diet immediately after feeding, unlike the other pesticides, which were dissolved directly within the diet.

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
