# Peer review of "In Vitro Effects of Pesticides on European Foulbrood in Honeybee Larvae"

_insects, 2020, doi:10.3390/insects11040252_

Round 1

Reviewer 1 Report

This article addresses the question of the involvement of certain pesticides in the appearance of clinical symptoms of European foulbrood in honey bees. This sanitary problem is currently marked in North America among beekeepers using their colonies for pollination of blueberries. The strain of bacteria used in the pesticide exposure trials was isolated in the field by the authors and is defined as an "atypical" strain. European foulbrood is a growing problem in the world and its epidemiology is still unknown and deserves more attention in order to reduce the use of antibiotics or the destruction of colonies. The authors present beautiful and informative histological sections of healthy and diseased larvae.

General remarks:
We know that the bacterium Melissococcus plutonius is highly variable genetically and phenotypically. The conclusions presented in the article are based on only one strain. The distribution and frequency of this strain in the field is unknown. For a good quality article, it would be necessary to confirm the observations with results for several other strains of the bacteria or to know the distribution of this strain in a region. An alternative would be to relativize the results by explaining that the observations apply only to this strain and that its distribution is not known. The article thus loses much of its relevance.
As the effects were only observed over 6 days, it cannot be excluded that similar effects could occur in the control groups but with a time lag. The observations must therefore be put into perspective in relation to the conditions tested.
A comparison with the existing literature concerning in vitro mortality with atypical strain contamination shows that the infections via the diet at D0 are much more higher than the infections used in this article (50, 250 and 500 CFU/larvae). Arai et al, (PlosOne, 2012) uses 50'000 CFU/larva and Nakamura et al, (Scientific reports, 2016) uses between 2'800 and 6'700 CFU/larvae. Please check your infection calculations and comment on the possible causes of this difference.
Why talk about highly and less virulent strains (Arai only talks about typical and atypical)? Nor does Nakamura say that atypical is systematically linked to high virulence. He only says that his tested strain of CC12 is more virulent than the other two strains of CC3 and CC13 (only one strain tested by CC). No rules can be drawn. In the test submitted, only one strain is tested and therefore there is no possibility of comparison.

Line 22, 87 and 324: replace “developed” with “used” (already developed by Schmehl et al., 2016 and Lewkowski et al., 2018)

Line 23-24: Must introduce in this sentence that the dosages tested are not field-relevant.

Line 29: “calculated field-relevant exposure”

Line 37: …leading to larval death especially under…

Line 47: you give the references (6-10) but McKee and Lewkowski did'nt test atypical strain

Line 48: agricultural and apicultural (Bogdanov, 2006, Apidologie, Contaminants of bee products)

Line 49: … insecticides through nectar, honeydew, honey, wax, pollen…

Line 56-65: Use the same unit for the all paragraph (ppm or preferably ng/g)

Line 76: add reference DOI: 10.1021/acs.est.8b01801, Christen et al., Environmental Science and Technology, 2018

Line 93: Why 0-12 hours? How do you recognize 0-12 hours old larvae if you caged the queen during 24 hours? (according line 119, queen caging during 24 hours)

Line 140: the brand and type of microscope are not necessary.

Line 143: … was required to have ≥ 75 % survival at D6 for the data from…..

Line 148: it is not clear to me whether these are true replicas or repetitions (several time-staggered series or several STCPs done simultaneously in the same series).

Line 162: What are these “positive controls”? The explanation is only available later, on line 171. Not logical.

Line 177 and 183: (2721 and 2,876) Use the same writing for the figures.

Line 194: Explain why you do that. e.g. “To determine the bacterial load of the stock culture,…

Line 196-198: Put this sentence after the chapter ending at line 203, after the explanation of the 3 infection groups.

Line 212: replace “survival from D0-6 with “survival at D6”

Line 213-215: The authors don't have to say for which group (50 CFU) they perform this supplemental test but in which case they would perform it. At this stage, we don't know why only for 50 CFU.

Line 213: Pearson Chi-squared test. At my opinion, a angular transformation is necessary before carrying out the Pearson Chi-squared test (when %-tage is between 0 to 30% or 70 to 100%)

Line 231, 237, 281: Why suddenly 7 and not 6 days?

Line 228-235: Comparing the survival rate for the 50 CFU group with the 1 ng/µl THI diet (Fig. 2A) and the PROP THI 1 ng/µl diet (Fig. 2D), it could be concluded that the addition of PROP improves survival. This needs to be discussed.

Line 242-243 and 263-265: Notes on the graph should be placed after the explanations. This sentence should be placed at the end of the legend. This remark also applies to figure 3.

Line 248: replace “over 7 days” with “at day 6”

Line 279-284: It is, in my opinion, difficult to make such a statement because only one of the 3 contaminations tested comes to this conclusion. There are certainly several replications, but it is not clear whether the authors made several independent series. Please also comment on the comparison with the diet THI only and 50 CFU (Fig. 2A).

Line 298-299: Please give a possible explanation for the decrease in survival at 50 CFU and not at 250 and 500 CFU.

Line 318: …that the combined stress of sublethal bacterial infection and sublethal pesticide exposure may increase…

Line 324: replace “highly virulent” with “atypical”

Line 328: …increase larval mortality from EFB at low dosage in vitro,…

Reviewer 2 Report

This is a well-written paper and an interesting read. I especially appreciate that the authors avoid to draw far-fetched conclusions based on few, slightly significant results and keep the discussion balanced.

Material & Methods

Line 125: Was the royal jelly free of or tested for chemicals?

Line 208: Were the larvae washed before culturing? If not, how do you know that the bacteria is not from the feed?

Results:

Line 236: THI is an insecticide, not a fungicide as stated in the text.

Figure 1 legend: Replace "reproduction of European foulbrood" with "multiplication of Melissococcus plutonius"

Figur 3: Try to make this figure clearer. Not enough difference between the legends makes it hard to read.

Reviewer 3 Report

The research has been prompted by the problem of EFB in bees pollinating blueberry fields in Canada, which has been known for decades [29]. The authors completed a laboratory experiment to detect if exposure to certain pesticides increases the risk of EFB in bees and gained some results which may (partially) ease the worries of beekeepers and blueberry growers.

However, there are some flaws that cannot be overcome: 

The research design is not appropriate: In fact, the concentration of thiamethoxam you tested is much higher than field-realistic. Pilling et al (2013) reported residues of up to 4 microgram thiamethoxam/kg in oilseed rape nectar; See also thiamethoxam concentrations that Coulon et al. (2019) refereed in the fifth paragraph of Introduction. “13.3 ng/g in nectar from oil-seed rape, and 86 ng/g in pollen from field margin plants [Botias et al., 2015]. It has also been detected in hive matrices, at a maximum of 20.2 ng/g in honey [Barganska et al., 2013], and 53.3 ng/g in stored pollen [Mullin et al., 2010].”

Note: When reviewing pollen residue data, the level of consumption by bees should be taken into account (<10% pollen in diet).

Note: nanogram/gram = microgram/kilogram

Thus, based on all available data, the most realistic concentration of thiamethoxam that may reach honey bee appears to be closer to or below 10 microgram/kilogram (10 ppb).

A concentration of 100 microgram/kilogram (100 ppb) seldom observed in field data and known to result in colony loss alone (Overmyer et al, 2017, Thompson et al, 2019).

You tested 1 and 10 nanogram/microliter = 1000 and 10,000 microgram/kilogram (or 1000 and 10,000 ppb) which is far away from realistic level that may reach honey bees.

Mentioned references:

  • Pilling E, Campbell P, Coulson M, Ruddle N, Tornier I (2013) A four-year field program investigating long-term effects of repeated exposure of honey bee colonies to flowering crops treated with thiamethoxam. PLoS One 8 (10) e77193.
  • Coulon M, Schurr F, Martel AC, Cougoule N, Begaud A, Mangoni P, Di Prisco G, Dalmon A, Alaux C, Ribiere-Chabert M, Le Conte Y (2019) Influence of chronic exposure to thiamethoxam and chronic bee paralysis virus on winter honey bees. PloS One 14 (8) e0220703.
  • Botıas C, David A, Horwood J, Abdul-Sada A, Nicholls E, Hill EM, et al (2015) Neonicotinoid residues in wildflowers, a potential route of chronic exposure for bees. Environmental Science & Technology 49 (21) 12731-12740.
  • Barganska Z, Slebioda M, Namiesnik J (2013) Pesticide residues levels in honey from apiaries located of Northern Poland. Food Control 31 (1) 196–201.
  • Mullin CA, Frazier M, Frazier JL, Ashcraft S, Simonds R, vanEngelsdorp D, et al (2010) H igh Levels of Miticides and Agrochemicals in North American Apiaries: Implications for Honey Bee Health. PLoS One 5 (3) e9754.
  • Overmyer J, Feken M, Ruddle N, Bocksch S, Hill M, Thompson H (2018) Thiamethoxam honey bee colony feeding study: Linking effects at the level of the individual to those at the colony level. Environmental Toxicology and Chemistry 37 (3) 816-828.
  • Thompson H, Overmyer J, Feken M, Ruddle N, Vaughan S, Scorgie E, Bocksch S, Hill M (2019) Thiamethoxam: Long-term effects following honey bee colony-level exposure and implications for risk assessment. Science of The Total Environment 654, 60-71.

To conclude: Incorrect concentrations of thiamethoxam alone are sufficient to reject the paper; Having seen these, I did not check the concentrations of other three pesticides, Boscalid (BOS), Pyrimethanil (PYR) and Propiconazole.

Line 21-22 - who pollinates blueberries? Beekeepers!? No, but bees.

Line 41 - Nothing can be “characterized into” something, BUT only characterized as, or classified as, or classified according to something.

Line 47 - “higher incidence of larval mortality relative to typical strains”!? Better IN COMPARISON WITH

Problematic usage of definite articles in the MS.

Line 73-74 – after “innate immune function” you shoud insert references, e. g.

  • Tesovnik T, Zorc M, Ristanić M, Glavinić U, Stevanović J, Narat M, Stanimirović Z (2020) Exposure of honey bee larvae to thiamethoxam and its interaction with Nosema ceranae infection in adult honey bees. Environmental Pollution 256, 113443
  • Glavinic U, Tesovnik T, Stevanovic J, Zorc M, Cizelj I, Stanimirovic Z, Narat M (2019) Response of adult honey bees treated in larval stage with prochloraz to infection with Nosema ceranae. PeerJ 7:e6325.

Line 104-105 – Why is MALDI-TOF mentioned here since it was not applied in ref [32]? It is necessary to describe the MALDI-TOF method or to refer to an appropriate reference. Finally, MALDI-TOF is not necessary if PCR is used. In addition, there is no evidence of M. plutonius identification using “Gram stain, MALDI-TOF mass spectrometry, and PCR”.

- It is necessary to provide some figures for the methodology explained in subsections 2.3, 2.4 and 2.5.

Line 137 - Not “From D2-5”, but “From D2 to D5!” And elsewhere.

Line 212 - “compared among”         It can be “compare within” OR “compared between”, regardless of the number of groups

Line 231 - “…had a significant, 25% lower survival over 7 days …”       I suppose it is “a significantly lower (by 25%) survival…”

Line 144, 229, 234, 240, 282 - “Infection control”      better, “infected control”

Line 293 - “…larvae… exposed the neonicotinoid clothianidin, experienced…” “…exposed to…”

- the outcome/result cannot be synergistic, but only the action of some factors (two or more) affecting some target, as it is correctly described in [2]

Line 310 - “…field relevant doses…” It is “…field-relevant…”

Line 318  - “… stress of sublethal bacterial infection and pesticide exposure…”   It is “… stressed posed by sublethal….” Stress is a reaction of an organism to some factors.

Line 331 - You cannot “further understand”, but only better or fully understand something.

Reviewer 4 Report

Thank you for the opportunity to review the manuscript titled, Effects of pesticides on mortality of Apis mellifera larvae from European foulbrood in vitro by Sarah C. Wood, et al.

Information for Authors and Editor:

Summary: This manuscript outlines a useful method of evaluating the effect of pesticide exposure on honey bee larvae. Ongoing concerns over pesticide usage and global loss of pollinators underscore the importance of this study. The manuscript is well organized and carefully presented.

Specific comments for Authors and Editor:

Particularly informative aspects of this study include:

  1. use of a highly virulent Canadian isolate of Melissococcus plutonius
  2. the ability to test pesticides individually and in combination (revealing synergistic effects) using an in vitro method
  3. evaluating outcome of exposure to living honey bee larvae

Major concern for Authors and Editor:

  1. An irrelevant bacterium should be tested in parallel with the M. plutonius pathogen. This important control would verify that effects of the highly virulent strain of M. plutonius examined are not caused by nonspecific variables. If an irrelevant bacterium is not used, rationale for testing the pathogen alone should be provided.

Minor concerns:

  1. To satisfy Koch’s postulates, disease or death in larvae infected with the M. plutonius should be specifically shown or stated. The appearance of microbes in the gut of larvae exposed to M. plutonius (Figure 1d) does not appear in itself to be evidence of pathology. If it is considered evidence of pathology, that should be clearly justified. Can comment be included to indicate the larvae represented in Figure 1b are dead or decreased in mass?
  2. The legend for Figure 1d indicates a Gram-stained tissue section is represented. However, it is not clear if the tissue section shown in Figure 1c represents H&E staining or Gram staining. The staining methods shown in the images should be clearly stated.
  3. It is not clear in Table 1 why the BOS diet concentration (29.25) is shown with four significant figures, while the other pesticides are shown in one or two significant figures.
  4. A brief rational for three different larvae diets (A, B, and C) would benefit the reader.

Round 2

Reviewer 1 Report

Thanks for the corrections that have greatly improved the manuscript.

Here some points:

Figure 1: It's a bit confusing that two groups have the same name (Pesticide-treated).
I am not convinced that the images at D0 are very informative, especially since the larva presented is at a much more advanced stage.
A table representation might be better with a first column "Inoculation by M. plutonius D0" and a second column "Pesticide administration D0 - D5".
We thus have a cross table with a more synthetic view.

Line 363: replace "reproduce" with "trigger"

Excel-File, tab labeled “CFU”: please, explain under this table how you calculate the dilution factor because the information provided in the article is not sufficient to understand this figure.

Reviewer 3 Report

I did not change my opinion that concentrations used are excessive. Indeed, the authors acknowledge it by repeating this fact several times in manuscript, but I wonder if this justifies such research?

E.g. in case of thiametoxam, the authors tested 1 and 10 nanogram/microliter = 1000 and 10,000 microgram/kilogram (or 1000 and 10,000 ppb), which is far beyond realistic levels that may reach honey bees (10 ppb - 100 ppb). This calculation is based on numerous references given in my previous reviewer report.

Based on authors' reply to my criticisms related to excessive concentrations, I have realized that the authors do not know the rules for selecting concentrations for toxicity testing and I would recommend to them to follow the guidelines summarized in:

Medrzycki P, Giffard H, Aupinel P, Belzunces LP, Chauzat MP, Claßen C, Colin ME, Dupont T, Girolami V, Johnson R, Le Conte Y et al. (2013) Standard methods for toxicology research in Apis mellifera. In V Dietemann; J D Ellis; P Neumann (Eds) The COLOSS BEEBOOK, Volume I: standard methods for Apis mellifera research. Journal of Apicultural Research 52(4): http://dx.doi.org/10.3896/IBRA.1.52.4.14

However, if the editor and the other reviewers think that this work may be published, I would just ask for correction of Abstract (line 23-24) as follows: the sentence “Using pesticide doses in excess or equal to maximal field-relevant concentrations, …” should be split in two. In first sentence, please precisely indicate which pesticides were analysed using excessive concentrations, which in maximal field-relevant concentrations and which ones in field-relevant concentrations. Considering the limit of words in Abstract, I propose writing (please check the accuracy), “Concentrations tested were excessive (thiamethoxam and pyrimethanil), maximal field-relevant (propiconazole) or field-relevant (boscalid)”.

Finally, the authors must know that the terms concentration and dose do not have the same meaning and cannot be used interchangeably. Their use in the manuscript has to be checked thoroughly and corrected accordingly.

Reviewer 4 Report

I have no further concerns. Thanks to the editors, and especially thanks to the authors for this research.

Author Response

Thank you.